# Highly Washable and Conductive Cotton E-textiles Based on Electrochemically Exfoliated Graphene

**DOI:** 10.3390/ma16030958

**Published:** 2023-01-19

**Authors:** Zakhar Ivanovich Evseev, Fedora Dmitrievna Vasileva, Svetlana Afanasyevna Smagulova, Petr Stanislavovich Dmitriev

**Affiliations:** Institute of Physics and Technologies, North-Eastern Federal University, 677000 Yakutsk, Russia

**Keywords:** graphene, graphene oxide, electrochemically exfoliated graphene, e-textiles, wearable electronics

## Abstract

In this study, cotton e-textiles were obtained using two types of graphene oxide. The first type of graphene oxide was synthesized using the Hummers’ method. The second type was obtained by the electrochemical exfoliation of graphite in an ammonium salt solution. It was shown that e-textiles based on electrochemically exfoliated graphene have a higher electrical conductivity (2 kΩ/sq) than e-textiles based on graphene oxide obtained by the Hummers’ method (585 kΩ/sq). In addition, textiles based on electrochemically exfoliated graphene exhibit better washing and mechanical stress stability. The electrical resistance of the e-textiles increased only 1.86 times after 10 cycles of washing, compared with 48 times for the Hummers’ method graphene oxide textiles. The X-ray photoelectron spectra of the two types of graphene oxides showed similarity in their functional compositions after reduction. Studies of individual graphene flakes by atomic force microscopy showed that graphene oxide of the second type had a smaller lateral size. Raman spectroscopy showed a higher degree of sp^2^ structure regeneration after reduction for the second type of graphene. These properties and the tendency to form agglomerated particles determine the mechanochemical stability and high electrical conductivity of e-textiles based on electrochemically exfoliated graphene.

## 1. Introduction

Recently, graphene and related materials have been increasingly applied in various fields [1,2,3,4]. One of the actively developing areas is the use of graphene in the field of e-textiles because it can be used to develop wearable and smart devices, such as body movement sensors [5], vital sign monitoring [6], electronic heaters [7], supercapacitors [8], and others [9]. The scientific community has particularly focused on graphene oxide (GO), owing to its unique properties [10,11,12]. GO can form stable aqueous suspensions, due to the presence of oxygen functional groups [13]. This simplifies many of the technological processes associated with the deposition of GO on textile surfaces. In addition, oxygen groups bind to GO flakes via van der Waals forces in many natural and artificial fabrics [14,15,16].

The main problem with e-textiles based on GO is their relatively high electrical resistance and low washing stability [17,18]. In most studies, the resistance of the obtained e-textiles is in the range of tens of kΩ/sq., which leads to the need for relatively high-voltage power supplies of approximately 20–50 V [19]. Such high voltages pose certain dangers to the wearer. Additionally, most wearable electronic products are designed for use at 3.3–12 V. To increase the conductivity of the GO film, it is necessary to carry out deep reduction [20]. However, this leads to a significant decrease in the concentration of functional groups [21] that are responsible for binding to textile macromolecules. This leads to the rapid deterioration of the GO film on the surface of the textile during mechanochemical processing, such as washing [14]. Another problem is that toxic reducing agents are generally used for deep reduction. For example, hydrazine hydrate [22] is undesirable for technological processes related to wearable clothing, owing to its high toxicity. Therefore, some studies have used high-temperature treatment of GO e-textiles [23]. This, on the one hand, leads to a significant increase in electrical conductivity, but, on the other end, causes thermal destruction of textile fibers and leads to a deterioration in the strength characteristics of the resulting products [24].

The above problems can be solved by using GO with a low oxygen content, thus eliminating the need for deep reduction. Many methods have been developed for the synthesis of GO with different compositions and concentrations of functional groups [25]. One of the most interesting methods is electrochemical exfoliation [26]. This method makes it possible to obtain mildly oxidized graphene (MOG) with a small content of functional groups quickly and in large quantities [27]. Because of its lower oxidation state compared to GO synthesized by the Hummers’ method, it can be used to produce highly conductive films.

Currently, there are only a few studies on the use of electrochemically exfoliated graphene (EEG) in e-textiles. In [28], Zapata-Hernandez et al. compared GO obtained using the Hummers’ method and EEG deposited on cotton. The use of EEG made it possible to obtain fabrics with outstanding conductivity (0.66 kΩ∙m), compared with reduced graphene oxide (29.5 kΩ∙m). However, in this study, GO was reduced before being applied to the fabric, whereas the EEG was not reduced. After reduction, GO exhibits low dispersibility and quickly agglomerates. This creates difficulties in conducting technological processes, due to the need for constant sonication. In addition, the mechanochemical stability of the obtained e-textiles was not evaluated. Yuon Kim et al. [29] used EEG deposited on cotton to create piezoresistive sensors. Using the thermal reduction of graphene applied to the fabric by a hot press, e-textiles with extremely high conductivity (1.3 Ω/sq) were obtained. However, as mentioned above, the use of heat treatment is undesirable because it leads to the thermal destruction of the fabric. In addition, the effect of mechanochemical processing was not investigated in this study. To the best of our knowledge, the use of EEG in the field of e-textiles has only been explored in these two studies.

The development of technologies for producing conductive e-textiles based on GO, without the use of toxic reducing agents and a high-temperature process, while ensuring the high washing stability of the GO films, remains an important problem. The effect of the synthesis methods and physicochemical properties of the obtained graphene materials on the mechanochemical stability of e-textiles is a poorly studied area. In particular, the mechanochemical stability of e-textiles based on EEG has not yet been investigated. In this study, we compared the chemical composition of GO obtained by the Hummers’ method and EEG. Samples of cotton e-textiles were obtained using a simple dip-coating technique and were reduced using non-toxic agent commonly used in the textile industry. The electrical conductivities and mechanochemical stabilities of the obtained e-textiles were compared. The possible application of this technology to wearable heating devices was demonstrated.

## 2. Materials and Methods

GO was synthesized using two methods. The first type of GO was synthesized by the modified Hummers’ method described in our previous study [30]. In contrast to the conventional Hummers’ method, the method used in this study does not involve the use of an ice bath or ultrasonic decomposition of graphite oxide, and intercalation is achieved by increasing the mixing time. The second type of GO was synthesized by electrochemical exfoliation of graphite in an aqueous solution of ammonium sulfate (NH_4_)_2_SO_4_ in combination with ultrasound, as described in our previous paper [27]. Briefly, electrochemical exfoliation was carried out in a laboratory glass vessel that contained a 0.1 M electrolyte. A gold foil was used as the cathode. An ESA-16 graphite electrode (Polyprof-L LLC, Moscow, Russia) was used as an anode. The graphite exfoliation reaction continued for 30 min at a voltage of 15 V between the electrodes. After the reaction, a solid precipitate was isolated using vacuum filtration on a polytetrafluoroethylene track membrane with a pore size of 0.2 μm. The precipitate was thoroughly washed with deionized water to remove residual ammonium sulfate. The dry residue was dissolved in 100 mL of deionized water and subjected to ultrasonic treatment, using an Up 200St homogenizer (Hielscher Ultrasonics, Teltow, Germany) at a power of 50 W for 30 min. Hereafter, the second type of GO is referred to as the MOG.

The suspensions were reduced using sodium dithionite (Na_2_S_2_O_4_), a common reducing agent used in the textile industry. The initial GO and MOG suspensions were diluted to a concentration of 0.75 g/L with deionized water. The concentrations of the suspensions were determined by weighing the dry residue on a Vibra XFR-125E analytical scale (Shinko Denshi, Tokyo, Japan). The resulting suspension (200 mL) was heated to 90 °C with constant stirring. Subsequently, 4 g of Na_2_S_2_O_4_ (Rushim, Moscow, Russia) was added to the heated suspensions. The mixture was stirred at constant temperature for 1 h. After the completion of the reduction process, the reduced GO (rGO) and MOG (rMOG) were extracted as black precipitates by vacuum filtration and washed thoroughly with deionized water.

To characterize the obtained materials, suspensions of GO and MOG were drop-cast onto an SiO_2_ substrate and dried at room temperature for 24 h. The rGO and rMOG were dried under room conditions for 24 h, and the resulting powder was poured into aluminum foil pots.

GO and MOG were studied by Raman spectroscopy with an Ntegra Spectra spectrometer (NT-MDT, Zelenograd, Russia), using a green laser with a wavelength of λ = 532 nm and an Andor Spectra grating with 600 lines/mm. To study the composition of the functional groups of the graphene films, infrared (IR) spectra were measured on a Varian 7000 FTIR (Varian, Palo Alto, CA, USA) in the range of 550–4000 cm^−1^, with a resolution of 8 cm^−1^. X-ray photoelectron spectroscopy (XPS) was used to study the quantitative composition of the functional groups. Measurements were performed on a SPECS photoelectron spectrometer (SPECS GmbH, Germany), using a PHOIBOS-150-MCD-9 hemispherical analyzer (MgK* radiation, h* = 1253.6 eV, 150 W). The binding energy scale was pre-calibrated to the positions of the Au_4_f7/2 (84.00 eV) and Cu_2_p3/2 (932.67 eV) island-level peaks. The rGO and rMOG samples were applied as fine powders to double-sided scotch. XPS spectra were recorded in 0.1 eV increments for state identification and quantitative analysis. The XPSPeak 4.1 program was used to decompose the C1s regions into individual spectral components. The morphology and thickness of the individual GO flakes were studied by atomic force microscopy (AFM) on Solver Next (NT-MDT, Zelenograd, Russia) in non-contact scanning mode.

The e-textile samples were fabricated in several steps (Figure 1). Initial pieces of bleached woven cotton calico with a density of 120 g/m^2^ (“Ivanovskaya sewing factory”, Ivanovo, Russia) were processed according to the technology of preparation for dyeing, which included rinsing, bleaching, and mercerization [31]. An aqueous suspension of GO and MOG with a concentration of 0.5 g/L was ultrasonicated in the ultrasonic bath Elmasonic S 40 H (Elma Schmidbauer GmbH, Singen, Germany) for 10 min. The textile samples were then immersed for 60 min in an aqueous suspension of GO or MOG at room temperature, under constant stirring. GO and MOG bound to the cotton fibers and formed a black film on the surface of the textile samples. Subsequently, the samples were again dried under normal conditions, until the applied layer was completely dry. GO and MOG were applied up to 10 times to increase the electrical conductivity of the obtained textiles. The samples were then reduced to an aqueous solution of Na_2_S_2_O_4_ with a concentration of 2% wt., at 90 °C, with constant stirring for one hour. To remove Na_2_S_2_O_4_ residues, the samples were cleaned in a 0.5% soap solution under constant stirring. After cleaning, the obtained e-textiles were washed in distilled water and dried under normal conditions, until completely dry. For electrical measurements, two electrical contacts in the form of strips made of silver paste were applied to the opposite areas of the samples at a distance of 15 mm from each other. The silver paste was then dried at room temperature for 24 h.

The current–voltage (C-V) characteristics were measured on an ASEC-03 electrical measurement unit (A.M. Prokhorov General Physics Institute of RAN, Moscow, Russia). To measure the change in the electrical resistance during bending, a mechanical device with two clamps and the ability to adjust the distance between them was used. To measure the electrical resistance from the temperature, the samples were placed on a thermal table and heated to 80 °C. Then, the samples were gradually cooled to −60 °C, with the measurement of the C-V characteristics in the −10 °C step. The surfaces of the samples were imaged using the scanning electron microscope (SEM) JEOL JSM-7800F (JEOL, Tokyo, Japan). To evaluate the mechanochemical durability, the samples were subjected to washing according to GOST 9733.4–83. The samples were placed in a washing solution that consisted of 0.5 g of soap, 0.2 g of soda and 100 mL of water. The detergent solution in the beaker that contained the samples was stirred using an electromechanical stirrer at 60 °C for 30 min at 300 rpm. Thermal images were obtained using a B1L thermal imager (HikMicro, Hangzhou, China).

## 3. Results and Discussion

Figure 2 shows the Raman spectra of GO and MOG. The spectra are typical for graphene oxide, with D and G peaks in the areas of 1350 and 1600 cm^−1^, respectively. The D peak is associated with the disordered crystal lattice and the formation of sp^3^ carbon, while the G peak is associated with the presence of the sp^2^ carbon lattice. The I(D)/I(G) peak intensity ratios were 0.99 for GO and 0.92 for MOG, which indicates a higher density of defects in GO [32]. The I(D)/I(G) ratio was 1.13 and 0.704 for rGO and rMOG, respectively. Additionally, the spectrum of rMOG shows a stronger 2D peak intensity compared to that of rGO, which indicates a higher degree of regeneration of the sp^2^ carbon bonds. The lateral sizes of the sp^2^ domains in the flakes of the graphene materials were estimated according to the method proposed by Cankado et al. [33]. Equation (1) was used for the estimation.
(1)$$L_{a}\left(\mathrm{nm}\right)=\left(2.4\cdot 10^{-10}\right)\lambda^{4}{\left(\frac{I_{D}}{I_{G}}\right)}^{-1}$$
where *L_a_* is the lateral size of the graphite crystallite following sp^2^ hybridization; *I_D_* and *I_G_* are the intensities of the *D* and *G* peaks in the Raman spectra; *λ* is the wavelength of excitation radiation (532 nm). The results are presented in Table 1. It can be observed that the size of the sp^2^ domains in the case of MOG has increased significantly after reduction, as confirmed by the enhancement of the intensity of the 2D peaks in the spectrum. The increase in the number of defects in GO may be due to the higher content of oxygen functional groups. During the reduction process, formation of gaseous products occurs [34], which can damage the GO film.

Figure 3 shows the IR spectra of the GO and MOG films before and after the reduction process. The IR spectra before reduction (Figure 3a) show typical peaks for graphene oxide, which match the previous data [35]. The spectrum shows an absorption maximum in the area of ~3200–3400 cm^−1^, corresponding to the vibrations of the hydroxyl groups. The peaks correspond to the valence vibrations of the C=O bonds of carboxyl groups (-COOH) on the edges of the layer planes in the area of 1723 cm^−1^, strain vibrations of the water molecules adsorbed on the GO surface at 1624 cm^−1^, valent vibrations of the C-O bonds of carboxyl groups at 1357 cm^−1^, valent vibrations of the C-O bonds of hydroxyl groups (1044 cm^−1^), and the C-O-C vibrations of the epoxy groups (978 cm^−1^). After reduction (Figure 3b), the absorption peaks in the area of 1728 cm^−1^ (-COOH), the peak in the area of 1627 cm^−1^ (C=C), the peak in the area of 1624 cm^−1^ (H_2_O), and the peak in the area of 978 cm^−1^ (C-O-C) remain, which indicate the incomplete reduction of rGO and rMOG.

Figure 4 shows the XPS spectra of GO and MOG before and after the reduction process. From the analysis of the binding energies of the spectral components presented in the C1s spectra after their decomposition into individual components, we can conclude that the peak with the binding energy value of 284.5 ± 0.1 eV is corresponds to graphite-like carbon (sp^2^ hybridization). Other peaks are corresponding to carbon in the following functional groups: 286.0 ± 0.2 for -C-O and C-O-C (epoxy and hydroxyl groups), 287.4 ± 0.2 for -C=O (carbonyl and ketone groups) and 288.6 ± 0.2 eV for -O=C-O and -COOH (carboxyl and ester groups). Table 2 shows the identification of the states and the Ci/C (sum.) ratios for each sample. The ratios obtained show that GO has a higher initial concentration of oxygen groups than MOG. After reduction, the concentrations of the oxygen groups and their functional compositions were almost the same. We can also conclude that Na_2_S_2_O_4_ has the strongest effect on the epoxy and hydroxyl groups, whereas it has almost no effect on the carboxyl and ester groups.

Figure 5 shows AFM images of the individual GO and MOG flakes. The average lateral sizes of the individual MOG flakes were approximately 0.1 μm, and their thicknesses ranged from 0.4 to 3 nm. The average lateral sizes of the flakes were ~1 μm and their thicknesses ranged from 0.4 to 2 nm for GO. This suggests that the individual MOG flakes obtained by electrochemical exfoliation have, on average, more graphene layers, but smaller lateral sizes compared with the GO obtained by the modified Hammers method.

Figure 6 shows SEM images of the e-textile surface after 1 and 10 cycles of GO and MOG deposition. It is shown that the two types of graphene oxide form films with different morphologies on the surface of the fibers. It can be observed that GO forms a more uniform film, while MOG forms a surface with a more developed structure and shows a tendency to create agglomerates on the surface of the textile. Figure 7 shows SEM images of the e-textile surface after 10 washing cycles. It can be observed that both types of graphene oxide on the surface of the fibers underwent significant washout relative to the original state. However, the MOG film was better preserved than the GO-based film.

Figure 8a shows the dependence of the electrical resistance of the samples on the number of coating cycles. It can be observed that the electrical resistance gradually decreases to 7–8 deposition cycles, after which saturation occurs, which is associated with the final formation of a conductive network. A further increase in graphene content did not lead to a significant increase in conductivity. The samples of e-textiles with GO exhibited dielectric properties prior to reduction. After reduction, the average resistance of the samples with GO was 585 kΩ/sq. The resistance of the samples with MOG before reduction was 280 MΩ/sq and about 2 kΩ/sq after reduction. Figure 8b shows the change in the electrical resistance of the samples relative to the initial resistance after the washing cycles. It can be observed that the samples based on MOG (change in electrical resistance by 1.86 times after 10 cycles) have a higher washing stability than the samples based on GO (change in electrical resistance by 48 times after 10 cycles).

Figure 9 shows the electrical resistance of the samples as a function of stretching/relaxation and bending. It can be observed that the samples based on rGO and rMOG have different patterns of electrical resistance change. rGO stretching leads to an irreversible increase in resistance, which is associated with the rupture of uniform films formed by GO on the surface of the fibers. For the samples based on rMOG, the reverse effect of a decrease in resistance was observed at small stretching values (up to 3 mm). This is possibly due to the compaction of the fit of the woven fabric individual nodes when it is stretched. This effect, however, to a lesser extent, can also be observed in fabrics based on rGO at stretch values of 1–2 mm. However, the destruction of rGO films eliminates its effect. In addition, after relaxation, the resistance value of the rMOG-based samples did not change significantly, which was associated with a more agglomerated structure of the rMOG coating that was not strongly affected by stretching. Mechanical bending tests were also performed with a radius of 3–4 mm. Measurements were performed before and after bending (Figure 7b). It can also be observed here that the uniformity of the rGO film contributes to the rapid degradation of the electrical network, due to the appearance of cracks under mechanical loads. Simultaneously, the conductivity of the samples based on rMOG improved, owing to the optimization of the distribution of agglomerated flakes on the surface of the fibers.

Based on the XPS data (Table 2), it is known that the compositions of the functional groups of rGO and rMOG after reduction are almost identical. The better electrical conductivity and resistance to washing shown by rMOG are associated with the morphological features of the rMOG film on the surface of the cotton fibers. It is believed that graphene oxides bind to the cellulose macromolecule through van der Waals interactions between the oxygen groups of GO and the hydroxyl groups of cellulose [14]. These interactions are considered to be weak and are easily disrupted. The AFM measurements showed that the obtained GO and MOG flakes consisted mainly of few-layered graphene. At the same time, owing to the small lateral dimensions of MOG flakes, these interactions exert a stronger force than in GO with large flakes, owing to the larger area-to-volume ratio of the former. Additionally, MOG, owing to the lower content of oxygen groups, has a lower hydrophilicity [36] and a greater tendency to form agglomerates on the fiber surface during coating, as shown in Figure 6. It is likely that these agglomerate structures can become stuck between the textile fibers (Figure 7), thus contributing to the higher mechanochemical stability of the rMOG films. The more homogeneous rGO film can easily undergo cracking under mechanical and mechanochemical stresses, which accounts for its higher electrical resistivity with the same functional composition as rMOG. This also occurred in the case of stretching and bending, as shown in Figure 9. The samples with rGO were subjected to irreversible deterioration in conductivity, owing to the cracking of the homogeneous film. Simultaneously, the rMOG samples even improved their conductivity, owing to the redistribution of agglomerates on the surface and between the fibers. In addition, it has been shown that the flakes in rMOG have domains with regenerated sp2 structures of larger sizes than those in rGO. This property causes the high electrical conductivity of rMOG-based e-textiles with the same functional composition as that of rGO. Further destruction of the rGO film during washing led to a significant increase in the electrical resistance of the sample. rMOG, due to its more developed surface morphology, preserves the electrically conductive chain under mechanochemical stress.

To demonstrate potential applications, the electrical heating performance and temperature sensitivity of the resulting conductive cotton samples were tested. The measurement results are shown in Figure 10. It can be observed that the electrical resistance is inversely proportional to the temperature. The sensitivity of the rGO sensors was higher than that of the rMOG sensors. This is due to the initially high resistance of rGO, since thermionic electrons have a much greater effect on its conductivity than in the case of rMOG with a relatively high electrical conductivity. Owing to its higher electrical conductivity, rMOG exhibited better heating capabilities than rGO (Figure 10b). When 10 V was applied, the heating rate of rMOG was 45 °C. rGO did not exhibit significant heating effects at voltages up to 30 V because of its high electrical resistance. Figure 9c shows the breathability of the fabric. An ultrasonic mist generator sealed with rMOG/cotton exhibited a small amount of vapor leakage. The breathability of the rMOG-coated fabric was comparable to that of untreated cotton in this demonstrative test. These properties demonstrate that rMOG-based e-textiles can be used as heating elements for clothing. Additionally, temperature sensitivity can help to implement built-in temperature control, which favorably affects user convenience.

## 4. Conclusions

Cotton e-textiles based on GO obtained using the Hummers’ method and electrochemical exfoliation have been studied. E-textiles based on electrochemically exfoliated graphene showed higher electrical conductivity (2 kΩ/sq vs. 585 kΩ/sq) and mechanochemical stability (change in electrical resistance by 1.86 times after 10 cycles vs. 48 times) than e-textiles based on GO obtained using the Hummers’ method. The data showed that one of the major factors in the mechanochemical stability of the graphene oxide films on the surface of cotton is the morphology of the films formed on the surface of the fibers. The smaller lateral size of the MOG flakes, as well as the more developed morphology of the films, contributed to improving the mechanochemical stability of the e-textiles. The development of methods for synthesizing GO with the required functional compositions and lateral dimensions is important for creating e-textiles with high electrical conductivity and mechanochemical stability. The MOG obtained by electrochemical exfoliation has great prospects for application in wearable electronics and e-textiles.

## Figures and Tables

**Figure 1 materials-16-00958-f001:**
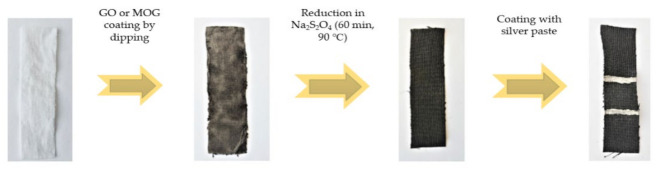
Stages of making samples of e-textiles.

**Figure 2 materials-16-00958-f002:**
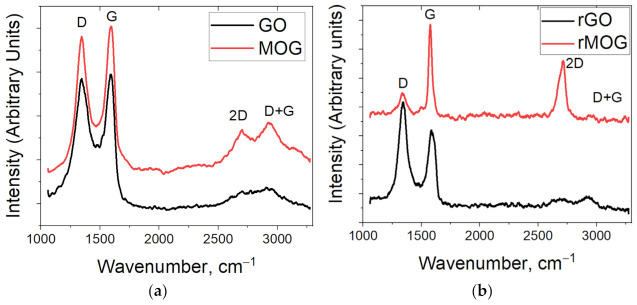
Raman spectra of the (**a**) GO and MOG films; (**b**) rGO and rMOG powders.

**Figure 3 materials-16-00958-f003:**
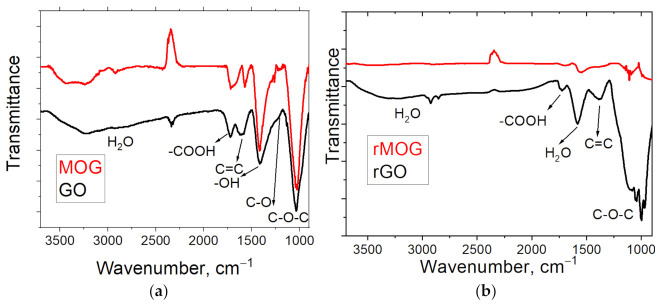
IR spectra of (**a**) GO and MOG films; (**b**) rGO and rMOG powders.

**Figure 4 materials-16-00958-f004:**
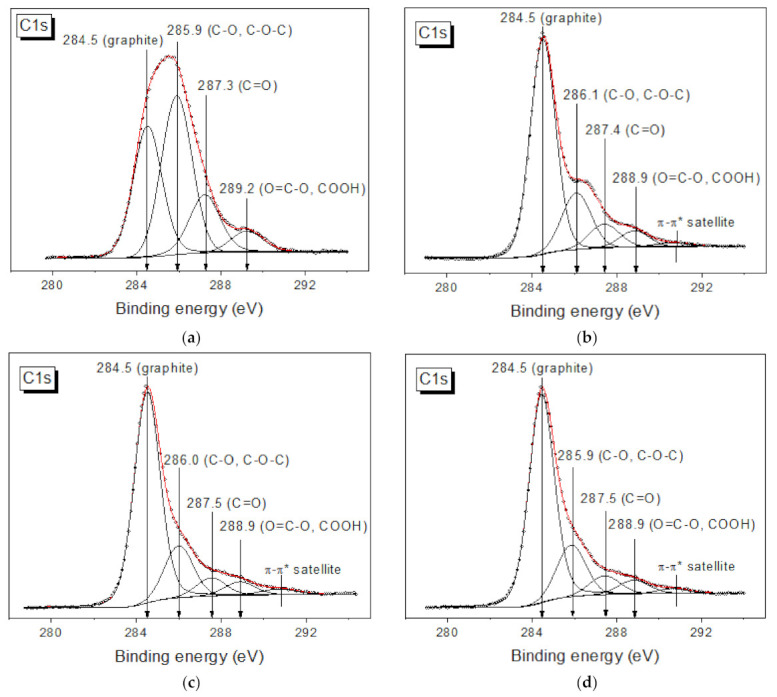
XPS spectra of (**a**) GO; (**b**) MOG; (**c**) rGO; (**d**) rMOG.

**Figure 5 materials-16-00958-f005:**
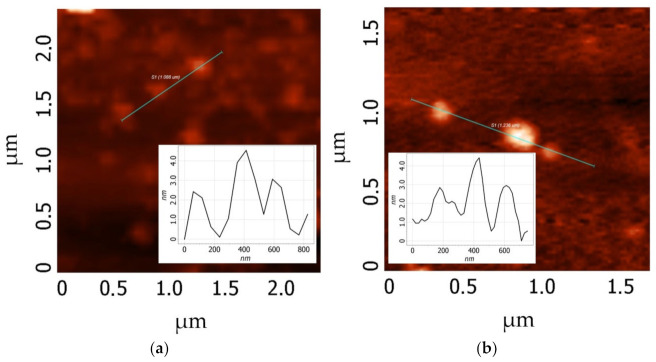
AFM images of individual flakes and their height profile: (**a**) GO; (**b**) MOG.

**Figure 6 materials-16-00958-f006:**
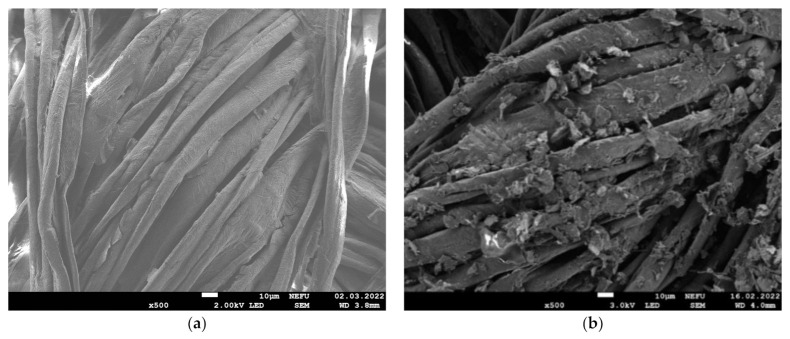
SEM images of the e-textile surface at 500× magnification: (**a**) GO after 1 coating cycle; (**b**) MOG after 1 coating cycle; (**c**) GO after 10 coating cycles; (**d**) MOG after 10 coating cycles.

**Figure 7 materials-16-00958-f007:**
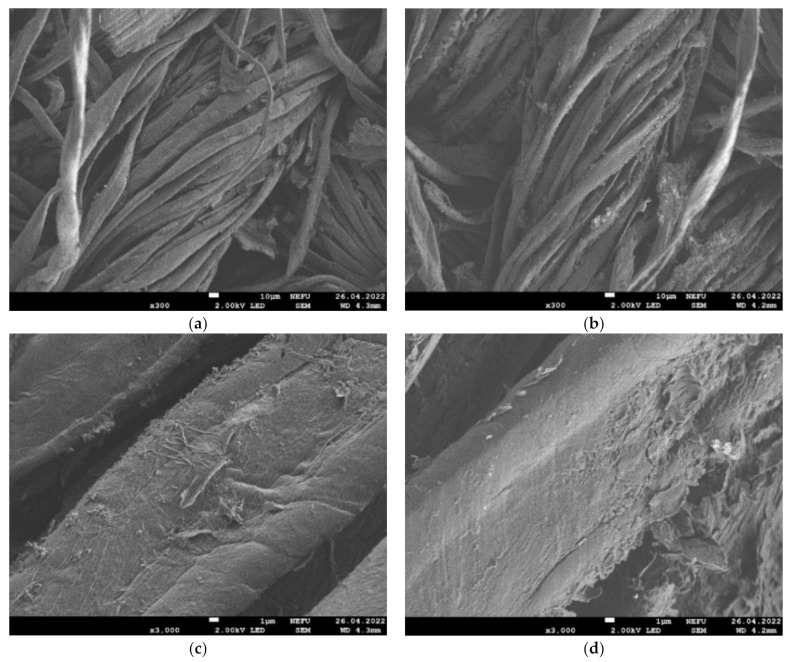
Image of e-textile surface after 10 washing cycles: (**a**) GO at 300× zoom; (**b**) MOG at 300× zoom; (**c**) GO at 3000× zoom; (**d**) MOG at 3000× zoom.

**Figure 8 materials-16-00958-f008:**
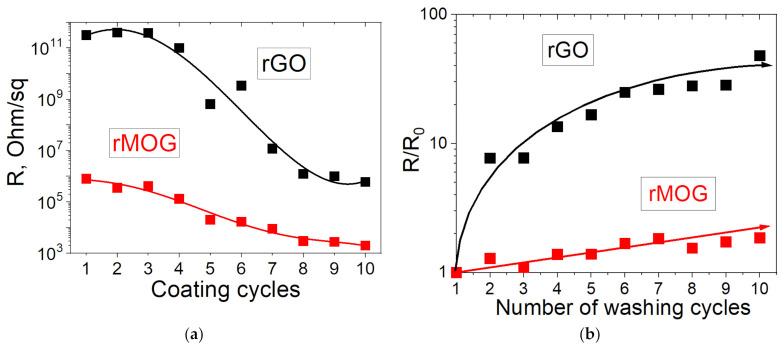
Change in the electrical resistance of fabrics with rGO and rMOG: (**a**) change from the coating cycles; (**b**) change from the number of washing cycles.

**Figure 9 materials-16-00958-f009:**
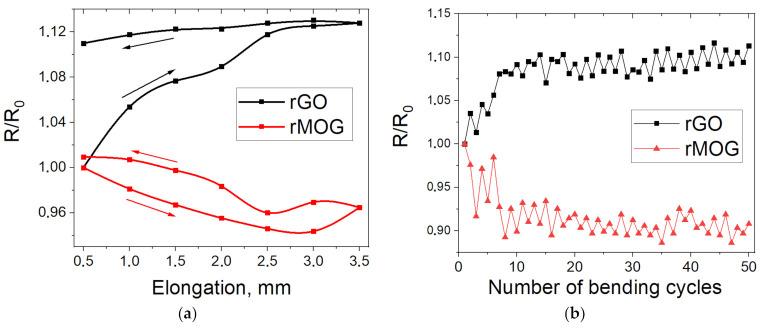
Changes in electrical resistance of rGO and rMOG e-textiles: (**a**) changes from elongation and subsequent relaxation; (**b**) changes from the number of bending cycles.

**Figure 10 materials-16-00958-f010:**
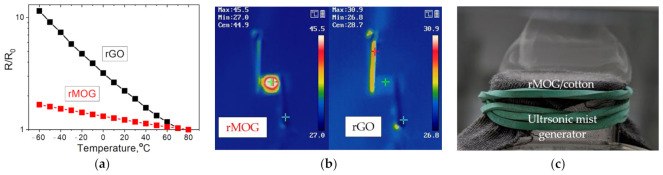
Demonstration of the practical application of rMOG e-textiles: (**a**) sensitivity of electrical resistance of rGO and rMOG to temperature; (**b**) demonstration of the heating performance of rGO and rMOG; (**c**) demonstration of rMOG/cotton breathability.

**Table 1 materials-16-00958-t001:** Lateral size of domains with sp^2^ hybridization.

Material	L_a_, nm
GO	36.22
MOG	21.81
rGO	33.49
rMOG	38.24

**Table 2 materials-16-00958-t002:** Contribution of carbon states to the total C1s spectrum (%).

Sample	C=C-%	C-O, C-O-C-%	C=O-%	O=C-O-%
GO	35.1	42.1	17.0	5.8
MOG	70.5	19.2	8.6	5.7
rGO	66.5	17.8	6.6	5.1
rMOG	70.0	18.0	7.0	5.0

## Data Availability

The data presented in this study are available upon request from the corresponding author. The data are not publicly available due to the data protection policy of the university.

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
