# Peer review of "Highly Washable and Conductive Cotton E-textiles Based on Electrochemically Exfoliated Graphene"

_materials, 2023, doi:10.3390/ma16030958_

Round 1
Reviewer 1 Report
Please check the file for details.

Author Response
Dear Reviewer,
Thank you for your review. Working on the comments and suggestions you made helped us improve the quality of manusctript.
Please check the attached file for responcse details.
Best regards,
authors

Reviewer 2 Report
Dear authors,
The data provided here for the article entitled "Highly washable and conductive e-textile based on electrochemically exfoliated graphene" are interesting; however, I am offering some comments throughout the manuscript.
(1) All short forms are not abbreviated. It is recommended to use abbreviations first and then continue in a short form.
(2) The introduction part is unprofessional; there is a lack of consistency between lines and paragraphs; it really needs to be revised very carefully.
(3) The main research gap of the work is totally absent in the Introduction. The last paragraph of the Introduction should provide information (only) about the science gap in the previous studies and what motivates you to do this review with the objective of the study.
(4) Fabric specifications (woven/knitted, yarn density) must be included.
(5) Results must be explained with proper reason.
(6) Figure 9 indicates the change of electrical resistance depending on temperature, but the authors have stated about the number of washing cycles.
(7) I am not an English speaker, but I found many typos and grammatical errors throughout the manuscript. These must be corrected and revised.
(8) References should be in accordance with the journal template.
Author Response
Deat Reviewer,
Thank you for your review. Working on the problems that you identified allowed us to improve our paper. It is great opportunity for us to receive feedback from the world scientific community.
Point 1: All short forms are not abbreviated. It is recommended to use abbreviations first and then continue in a short form.
Response 1: We revised all the abbreviations and corrected the identified shortcomings. The lines where abbreviations are entered are as follows:
Graphene oxide (GO) – line 31
Mildly oxidized graphene (MOG) – line 56
Electrochemically exfoliated graphene (EEG) – line 61
Reduced GO (rGO) and reduced MOG (rMOG) – line 112
Infrared (IR) – line 121
X-ray photoelectron spec-troscopy (XPS) – line 123
Atomic force microscopy (AFM) – line 132
Current–voltage (C-V) – line 153
Scanning electron microscope (SEM) – line 160
Point 2: The introduction part is unprofessional; there is a lack of consistency between lines and paragraphs; it really needs to be revised very carefully.
Response 2: Thank you for pointing out this flaw in our article. We have significantly revised the introduction of the article. Please check revised version of the article.
Point 3: The main research gap of the work is totally absent in the Introduction. The last paragraph of the Introduction should provide information (only) about the science gap in the previous studies and what motivates you to do this review with the objective of the study.
Response 3: Thanks for pointing out this problem. We have significantly revised the introduction of the article. Please check revised version of the article.
Point 4: Fabric specifications (woven/knitted, yarn density) must be included.
Response 4: We have included specification in lines 134-135:
Initial pieces of bleached woven cotton calico with a density of 120 g/m2 (“Ivanovskaya sewing factory”, Russia)
Point 5: Results must be explained with proper reason.
Response 5: We performed additional stretching/bending experiments to confirm the assumptions made in the discussion. The results are included in Figure 9. Added discussion of these results in lines 248-264.
Point 6: Figure 9 indicates the change of electrical resistance depending on temperature, but the authors have stated about the number of washing cycles.
Response 6: Thanks for pointing out the error, we've fixed it.
Point 7: I am not an English speaker, but I found many typos and grammatical errors throughout the manuscript. These must be corrected and revised.
Response 7: We have significantly revised the English of our manusctript. Please check revised version of the article.
Point 8: References should be in accordance with the journal template.
Response 8: We have revised this part, thank you for pointing out.
Best regards,
authors

Reviewer 3 Report
The authors reported the fabrication of cotton-based e-textiles from two types of graphene oxide (GO) synthesized by the widely used Hummers and electrochemically exfoliation method, and they showed that electrochemically exfoliated graphene have higher electrical conductivity (2 kΩ/sq) than e-textiles based on graphene oxide obtained by the Hummers method, as revealed by various characterizations including Raman spectroscopy, XPS and AFM. This manuscript can be accepted after minor revisions.
Comments and suggestions to justify my decision are as follows:
1. The more details of synthesis of GO via the two methods can be added in the main text.
2. More insightful discussion about change of electrical resistance of samples during cycles (Figure 6) should be provided.
3. Some relevant papers about GO-based materials electrochemical applications like Materials Today Sustainability, 2022, 17, 100096.; Electrochimica Acta 2015, 152, 216.; ChemSusChem 2013, 6, 474.; are suggested to be cited in the main text.
Author Response
Dear Reviewer,
Thank you for your review. It is great opportunity for us to receive feedback from the world scientific community.
Point 1: The more details of synthesis of GO via the two methods can be added in the main text.
Response 1: We added details in lines 90-93:
In contrast to the conventional Hummer’s method, the method used in this study does not involve the use of an ice bath or ultrasonic decomposition of graphite oxide, and intercalation is achieved by increasing the mixing time.
Also in lines 95-104:
Briefly, electrochemical exfoliation was carried out in a laboratory glass vessel containing a 0.1 M electrolyte. A gold foil was used as the cathode. An ESA-16 graphite electrode (Polyprof-L LLC, Russia) was used as anode. The graphite exfoliation reaction was continued for 30 min at a voltage of 15 V between the electrodes. After the reaction, a solid precipitate was isolated using vacuum filtration on a polytetrafluoroethylene track membrane with a pore size of 0.2 μm. The precipitate was thoroughly washed with deionized water to remove residual ammonium sulfate. The dry residue was dissolved in 100 ml of deionized water and subjected to ultrasonic treatment using an Up 200St homogenizer (Hielscher Ultrasonics, Germany) at a power of 50 W for 30 min.
Point 2: More insightful discussion about change of electrical resistance of samples during cycles (Figure 6) should be provided.
Response 2: We performed additional stretching/bending experiments to confirm the assumptions made in the discussion of resistance to mechanochemical processing. The results are included in Figure 9. Added discussion of these results in lines 248-264.
Point 3: Some relevant papers about GO-based materials electrochemical applications like Materials Today Sustainability, 2022, 17, 100096.; Electrochimica Acta 2015, 152, 216.; ChemSusChem 2013, 6, 474.; are suggested to be cited in the main text.
Response 3: Thank you for suggestion. We included these papers in introduction part, in lines 28 and 30.
Sincerely,
authors

Round 2
Reviewer 1 Report
The authors have properly addressed my concerns and it can be published now. Just a kind suggestion that might be helpful for authors' future study: when performing the breathability test, you can simply measure the residual weight of the water contained in the bottle by placing it in an ambient environment for several days.